**Data Availability Statement:** Data are available on https://datadryad.org/stash/share/kNWzQQ4egnoLIymWwJ2xmOPFi0mS0qIh9lghKdZFmsE.

# A harm reduction model for environmental tobacco smoke exposure among Bangladeshi rural household children: A modified Delphi technique approach

Rishad Choudhury Robin[1,2], Narongsak Noosorn[1]*

**1** Faculty of Public Health, Naresuan University, Phitsanulok, Thailand, **2** Shomman Foundation, Dhaka, Bangladesh

☺ These authors contributed equally to this work.

* nnoosorn@yahoo.com

## Abstract

This paper aimed to develop a harm reduction model to reduce exposure to environmental tobacco smoke among children of rural households in Bangladesh. A mixed-methods exploratory sequential design has been applied, and data has been collated from six randomly selected villages of Munshigonj district, Bangladesh. The research was divided into three phases. In the first phase, the problem was identified through key informant interviews and a cross-sectional study. In the second phase, the model was developed by focus group discussion, and in the third phase, the model was evaluated through the modified Delphi technique. The data was analyzed by thematic analysis and multivariate logistic regression in phase one, qualitative content analysis for phase two, and descriptive statistics in phase three. The key informant interviews showed attitude toward environmental tobacco smoke, lack of awareness, inadequate knowledge as a reason and smoke-free rules, religious beliefs, social norms, and social awareness as preclusion of environmental tobacco smoke. The cross-sectional study detected that households with no smoker (OR 0.006, 95% CI 0.002–0.021), high implantation of smoke-free household rules (OR 0.005, 95% CI 0.001–0.058), moderate (OR 0.045, 95% CI 0.004–0.461) to strong (OR 0.023, 95% CI 0.002–0.224) influence of social norm and culture along with neutral (OR 0.024, 95% CI 0.001–0.510) and positive (OR 0.029, 95% CI 0.001–0.561) peer pressure had been significantly associated with environmental tobacco smoke exposure. The final components of the harm reduction model consist of a smoke-free household, social norms and culture, peer support, social awareness and religious practice identified by the FGDs and modified Delphi technique.

## Introduction

Despite many efforts to reduce environmental tobacco smoke (ETS) exposure, around 890,000 deaths occur due to ETS yearly [1]. Children are the most susceptible population to ETS

**Funding:** The authors received no specific funding for this work.

**Competing interests:** The authors have declared that no competing interests exist

exposure, and involuntarily exposure to ETS is causing them to increase the risk for respiratory illness, cancer, allergy, and sudden infant death syndrome [2–4].

In developed countries, ETS exposure among children is high, indicating 89% in Turkey, 43% in Australia, 41% in the United Kingdom, and 33% in Canada [5]. However, the ETS prevalence in underdeveloped countries is also in the same line, 39.0% in Bangladesh, 38.7%, and 39.1% in India and Myanmar, respectively [6–8].

The high prevalence of ETS exposure in Bangladesh results from the low-cost availability of cigarettes (1). Moreover, most Bangladeshi people's perception reflected that ETS does not have any adverse effects, including male attitudes and behaviours that pose a significant barrier for reducing ETS exposure [9].

In Bangladesh, there is a scarcity of research on ETS. We have found only a few studies on ETS in Bangladesh, and none of them focused on the exposure of children in a rural households, which imitated a significant knowledge gap of crucial public health issues. To replenish the knowledge gap, we aimed to develop a harm reduction model to reduce exposure to ETS among rural area's household children of Bangladesh, thus helping to reduce the overall exposure of ETS around the country.

## Methods and material

### Research design and place

A mixed-methods exploratory sequential design was carried out between July 2018 and April 2019. The study was divided into three phases. In the first phase, the tobacco problem of the study area was identified, the model was developed in the second phase, and the model was evaluated in the last phase (Fig 1). Munshiganj district had been selected as the representative of the rural area from the 61 same cultural and ethnic background districts of Bangladesh through the lottery method of simple random sampling. Though there are 64 districts, three are from hill tracks that were not included in the lottery as the ethnic background of the populations is different from the rest of the 61 districts [10, 11]. The study was approved by the Institutional Review Board of Naresuan University, Thailand (COA No. 675/2018, IRB No.0502/61), and all the participants gave written consent before participating in the study.

### Phase 1—Identify problem

To identify the situation of ETS exposure, both qualitative and quantitative methods were used to obtain an overview of the situation.

**A. Key informant interview.** In-depth interviews were conducted to determine the respondents' behavior patterns regarding ETS exposure in the community. A key informant interview (KII) was used as the research tool for KII; active adult smokers, non-smokers, local political leaders, local government officials, law enforcement officials, health-related officials, religious leaders, teachers, lawyers, and shopkeepers were selected. The KII generated necessary information on the respondents' ETS to identify the variables and develop a different quantitative method questionnaire. It helped to understand how stakeholders behave and react regarding ETS exposure in the community, community norms of smoking, government rules, implementation, and other associated factors. Data were collected until data saturation occurred, and a total number of 14 KIIs were done. A purposive sampling method was used to select the participants. To understand the qualitative results of the research, a thematic analysis was used [12].

**B. Cross-sectional study.** A cross-sectional study was conducted for this part. By applying the Cochran formula, the final respondents were 410 adult males and females, including smokers and non-smokers [13]. Simple random sampling was carried out to select the six villages among the Munshiganj district, and through systematic random sampling, every third

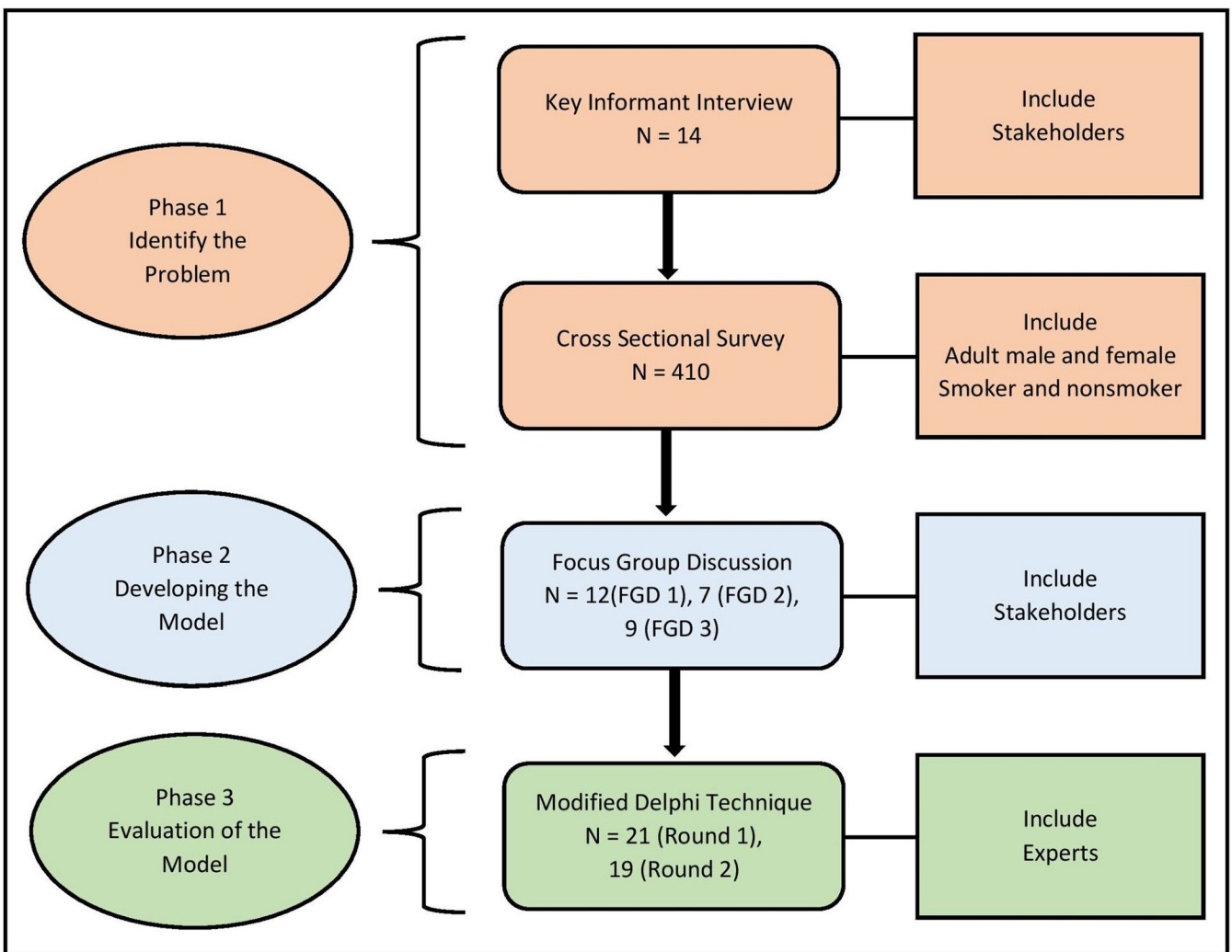

**Fig 1. Research diagram.**

household member was chosen as a sample. Population proportion had been calculated to select the number of representatives from each village. A self-administrative close-ended survey questionnaire on demographics, smoking status, self-reported exposures in the household, variables related to second-hand smoking exposure, and four constructs from the theory of planned behaviour (TPB) were chosen as a research tool. The study measured the ETS exposure on children from adult self-report [9, 14]. Exposure to passive smoking indicates as exposure to another person's tobacco smoke in the household for at least 15 minutes daily for more than one day every week in the past 30 days [15]. The questionnaire was developed in English and translated into Bengali (The national language of Bangladesh). Back translation was employed to ensure items' meaning and substance were not lost. SPSS version 20 software was used to analyze the data. Descriptive statistics were used to describe essential socio-demographic characteristics, whereas univariate and multivariate logistic regression models were used to investigate the association between independent and dependent variables. All results were presented as unadjusted and adjusted odds ratio (OR) with 95% confidence intervals (CIs). A p-value ≤0.05 was considered to be statistically significant.

### Phase 2—Developing the model

A proposed model was developed from the findings of phase one, focus group discussions (FGD), and guidelines from previous literature. A purposive sampling method was used to select the participants. Three heterogeneous FGDs had collected data. The participants include local political leaders, local government officials, law enforcement officials, health-related officials, religious leaders, teachers, shopkeepers, smokers, and non-smokers. The participant number for each group was 12, seven, and nine, respectively, and the duration of the discussion was 40 to 50 minutes. All FGDs were audiotaped and transcribed verbatim; the Principal Investigator (PI) checked the transcripts for consistency then a directed content analysis approach was conducted to analyze the data. This approach can forecast the variables of interest or the interactions between variables, which help define the initial coding system or associations between codes [16].

### Phase 3—Evaluation of the model

A close-ended online questionnaire was used as a tool where the tobacco field experts acknowledged the finding and gave their suggestions to evaluate the proposed model. Purposive sampling applied to select the experts. The experts consisted of researchers, doctors, nurses, psychiatrists, addiction specialists, social workers, and government and non-government officials responsible for tobacco control. A modified Delphi technique was used for data collection. Seventeen experts' opinions were needed to reduce the group error of less than 0.5 [17]. The modified Delphi technique is similar to the full Delphi technique. The major adjustment involves beginning the procedure with a set of prudently selected items. These preselected items may be drawn from several sources. The key advantages of this modification to the Delphi is that it is typically improves the initial round response rate, and provides a solid foundation in previously developed work [18]. Following these criteria, 21 experts participated in the first round and 18 in the second round. In the first round, a close-ended questionnaire developed from the finding of phase two was sent to the experts through email. The experts rated on a clearly defined nine-point Likert scale. In the second round, those items that did not achieve the consensus in the first round were again sent to the experts, informing the result from the first round together with the individual expert's rating and the median rating from the entire panel. In both rounds, the free-text responses were selected to represent their opinion's deepness. To calculate the strength of the consensus Interquartile range (IQR) is used. IQR is the absolute value of the difference between the 75th and 25th percentiles, with smaller values indicating higher degrees of consensus. The consensus was reached when one particular item IQR was $\leq 1$, and $\geq 75\%$ of the total items had IQR $\leq 1$ [19, 20]. Data were analyzed with the average percentage, mean, median, and IQR.

## Results

### Demographic characteristics

The demographic characteristics of the overall participants in phases 1 and 2 are presented in Table 1. In the KII, most of the 78.6% were male, and the participants were between 25 to 44 years. Among them, 85.8% were married and completed a degree from a university. Besides, almost one-third of the participants were non-smokers (35.7%). In the cross-sectional study, it was indicated an overwhelming participant was male (70.7%) majority were married (85.6%) and finished university study (37.5%). Almost one-fourth (26.6%) of the responders were smokers, also indicated by our study. In the FGD, male participants were maximum in number (64.2%). However, an equal number of participants in the age group 25 to 44 years and

**Table 1. Demographic characteristics of the participants.**

| Phase 1 | | | | | | Phase 2 | |
|---|---|---|---|---|---|---|---|
| Key informant interview | | Cross sectional survey | | | | Focus group discussion | |
| Characteristics | n = 14 (%) | Characteristics | n = 410 (%) | | | Characteristics | n = 28 (%) |
| **Gender** | | **Gender** | | | | **Gender** | |
| Male | 11 (78.6) | Male | 290 (70.7) | | | Male | 18 (64.2) |
| Female | 3 (21.4) | Female | 120 (29.3) | | | Female | 10 (35.7) |
| **Age (Years)** | | **Age (Years)** | | | | **Age (Years)** | |
| 18–24 | 1 (7.1) | 18–24 | 38 (9.3) | | | 18–24 | 8 (28.5) |
| 25–44 | 10 (71.4) | 25–44 | 205 (50.0) | | | 25–44 | 10 (35.7) |
| 45+ | 3 (21.5) | 45+ | 167 (40.7) | | | 45+ | 10 (35.7) |
| **Marital status** | | **Marital status** | | | | **Marital status** | |
| Single | 2 (14.2) | Single | 55 (13.4) | | | Single | 6 (21.4) |
| Married | 12 (85.8) | Married | 351 (85.6) | | | Married | 22 (78.5) |
| **Education** | | Divorce and Widow | 4 (1.0) | | | **Education** | |
| College | 2 (14.2) | **Education** | | | | Primary/Secondary | 8 (28.5) |
| University | 12 (85.8) | No formal schooling | 21 (5.1) | | | College | 12 (42.8) |
| **Employment** | | Primary schooling | 82 (20.0) | | | University | 8 (28.5) |
| Service holder | 7 (50.0) | Secondary schooling | 153 (37.4) | | | **Employment** | |
| Business | 5 (35.7) | University | 154 (37.5) | | | Service holder | 5 (17.8) |
| Unemployed | 2 (14.3) | **Employment** | | | | Business | 12 (42.8) |
| **Smoking Status** | | Service holder | 141 (34.4) | | | Unemployed | 11 (39.2) |
| Smoker | 5 (35.7) | Business | 157 (38.3) | | | **Smoking Status** | |
| Non-smoker | 9 (64.3) | Unemployed | 112 (27.3) | | | Smoker | 7 (25.0) |
| - | - | **Smoking Status** | | | | Non-smoker | 21 (75.0) |
| - | - | Smoker | 109 (26.6) | | | - | - |
| - | - | Non-smoker | 301 (73.4) | | | - | - |

more than 45 years (35.7%). In phase 1, the majority completed from university; however, in step 2, participants mostly completed their college (42.8%). In this phase, the number of smokers was found precisely one-fourth of the total respondents (25.0%).

## Phase 1—Identify the problem

In the thematic analysis, the core theme had been divided into reason and preclusion of ETS. ETS's cause was again codded as ETS's attitude, lack of awareness, and inadequate knowledge. A positive attitude can help to practice the behaviour while, on the opposite, a negative attitude can significantly impact the same reaction. For example, "Smoker is everywhere from young people to older people, everyone smokes. In the local market, stalls sell only cigarettes and teas, and you can see most people gathered in those stalls. People do not think it is a bad habit. They take it regularly like consuming an everyday meal (KII 103, School teacher of one of the unions)". People have inadequate knowledge of ETS exposure. They do not understand how a smoker's smoke impacts the people surrounding him. For example, one participant stated, "When a patient came to us mainly children with respiratory issues, if we ask if the father is a smoker or not, many times they get surprised by the question as they wonder what could be the relation between the parents smoking habit and the respiratory problem of their children (KII 108, Doctor of sub-district health complex). General people take smoking casually and are unaware of the consequence, "Smoking cigarettes is a regular practice, like taking meals

three times. They do not think about the harmful side effect of it. They do not think it affects him and others (KII 101, Religious leader from one of the local mosques)".

The preclusion of ETS had been codded as smoke-free rules, religious beliefs, social norms, and social awareness. Smoke-free laws indicated ETS-free households. Participants stated, "Protest should come from the home first. If the cigarette is not permitted inside the home, it eventually reduces the exposure (KII 110, Government official of Sub-district government office)". Furthermore, due to the modernization of society, children remain home for the majority of their time. As an example, "Time has changed. Nowadays, even the poorest family has a smartphone. You can get a Chinese smartphone even within 2500 taka. Children are now devoted to playing mobile games. That's made them stay at home. As a reason, indoor smoking causes more exposure to children. You cannot reduce it without banning smoking inside the house. In my opinion, this is the only salutation (KII 103, School Teacher of one of the unions)". A social norm is usually a reflection form religion and culture, and it has a significant influence on the human life of that particular society. In rural areas, females typically do not interfere with the husband's opinions. However, if they are adequately empowered, they can help to reduce ETS. One female participant explained, "We women can help. Yes, I admit that it is difficult in our society, but we can prevent our spouse from smoking inside the house if we have the empowerment. I know it will be challenging for some families but not impossible. Everyone loves his children, and no one wants to hurt them (KII 105, Smoker's wife from one of the selected villages)". As tobacco products are readily available, and there is no barrier to buying, it seems like a regular daily life activity. Local shopkeepers, mainly those who sell tea and cigarettes, are among the most familiar people in the rural area. Most of the adult males of the village went to their stall for evening tea. They can help with awareness buildup. The participant's opinion reflected that "People know each other in the village. Everyone knows who has a child or not. If the shopkeeper is aware of this effect, he can tell the father not to smoke inside the house or in front of children while buying cigarettes. People usually gossip in front of this type of cigarette shop. Shopkeepers have a good bonding with their customer. It's not liked the town. Everyone belongs to the same society (KII 101, Religious leader from one of the local mosques)". Religion has a tremendous influence on human life. It also influences the culture, norms, and different customary practices. Islam is the religion followed by the majority of the people in Bangladesh. Islamic way of life can reduce the exposure reflected by participant's opinion, "Obeying religious instructions can reduce smoking, and I also think passive smoking exposure like Bangladesh is a Muslim country with many smokers. If you ask any Muslim, they will reply smoking is prohibited in Islam. But still, they smoke. If they obey the Islamic way of life, I think it will reduce (KII 113, Female non-smoker from one of the selected villages)".

Moreover, our research observed that households with no smoker had significantly associated ETS exposure in both univariate (OR 0.014, 95% CI 0.007–0.028) and multivariate (OR 0.006, 95% CI 0.002–0.021) analysis. Similarly, high implantation of smoke-free rules in the household was also significant in univariate (OR 0.006, 95% CI 0.001–0.046) and multivariate analysis (OR 0.005, 95% CI 0.001–0.058). Additionally, moderate (OR 0.045, 95% CI 0.004–0.461) to strong (OR 0.023, 95% CI 0.002–0.224) influence of social norm and culture along with neutral (OR 0.024, 95% CI 0.001–0.510) and positive (OR 0.029, 95% CI 0.001–0.561) peer pressure had been significantly associating in the multivariate analysis (Table 2) The Hosmer and Lemeshow test was not significant ($x2$ = 7.481, df. = 8, p = 0.486), confirming the model's goodness of fit.

## Phase 2—Developing the model

According to the respondents, five themes derived in this phase from the content analysis were a. smoke-free household, b. Social norm and culture, c. Peer support, d. Social awareness and, e. Religious practice. The quotes representing each category are presented in Table 3.

**Table 2. Univariate and multivariate analysis to find out the association of independent and dependent variables.**

| Variables | n = 410 (%) | Unadjusted | | Adjusted* | |
|---|---|---|---|---|---|
| | | OR (95%CI) | P-value | OR (95%CI) | P-value |
| **Smoker presence in the household** | | | | | |
| Yes | 171 (41.70) | Reference | | | |
| No | 238 (58.04) | 0.014 (0.007–0.028) | **0.000** | 0.006 (0.002–0.021) | **0.000** |
| **Smoke-free rules in the household** | | | | | |
| Low implementation | 58 (14.14) | Reference | | | |
| High implementation | 351 (85.60) | 0.006 (0.001–0.046) | **0.000** | 0.005 (0.001–0.058) | **0.000** |
| **Social norm and culture** | | | | | |
| Low influence | 19 (4.63) | Reference | | | |
| Moderate influence | 86 (20.97) | 0.966 (0.357–2.614) | 0.946 | 0.045 (0.004–0.461) | **0.009** |
| High influence | 304 (74.14) | 0.550 (0.217–1.396) | 0.209 | 0.023 (0.002–0.224) | **0.001** |
| **Peer pressure** | | | | | |
| Negative | 310 (75.6) | Reference | | | |
| Neutral | 93 (22.7) | 1.652 (0.304–8.966) | 0.561 | 0.024 (0.001–0.510) | **0.017** |
| Positive | 7 (1.7) | 1.394 (0.266–7.306) | 0.694 | 0.029 (0.001–0.561) | **0.019** |

*Adjusted to age, gender, income, religious belief, knowledge, environmental impact, media influence, perception, attitude, intention, perceived behaviour control

**A. Smoke-free household.** A smoke-free household is an effective way to reduce ETS, according to the participants of FGDs. Home can act as an administrative body. A person got his first lesson from his home. Additionally, motivation and self-understanding are also necessary to change any behaviour (Quotes SFH 1–2). Making a specific place for smoking may have some pros and cons. However, the advantage of this is high with some proper regulation. Only creating a place for tobacco will not benefit without these regulations (Quotes SFH 3). Children can play an important role in making a house smoke-free and the other family member. Child well-being is the most important thing for any parent, helping him make the house smoke-free (Quotes SFH 4).

## B. Social norm and culture

A different norm and culture guide every society. The norm and culture have a significant influence on a person's behaviour. Social smoking is a common practice in Bangladesh. Similarly, people have minimal awareness regarding ETS. The social smoking practice causes more harm, and this customary should change for exposure reduction (Quotes SNC 1). In the rural area of Bangladesh, a woman does not enjoy the freedom of rights. Their husband dominates them. Even they are unable to ask their husband to stop smoking. However, a spouse can be one of the main support systems to reduce smoking and ETS. This scenario needs to be changed, and the government is trying hard to empower women. The government, social welfare division, helps women with microcredit loans, which eventually allows women and helps them get their rights to speak and make their family environment pollution-free (Quotes SNC 2–3).

## C. Peer support

Support from close family and friends consider a significant way to reduce ETS. Spouse support is essential to minimize exposure. Additionally, grandparents are also a significant cause of exposure. Their awareness is necessary to make the house exposure-free (Quotes PS 1). Children are the most precious in any human's life. Their knowledge and understanding can have

**Table 3. Selected quotes representing exposure of passive smoking among children.**

| Theme | Quotes |
|---|---|
| 1. Smoke-free household | SFH 1 - "Everything should start at home. It has more effect on isolated places rather than open spaces. A smoker should understand I can harm myself, but I cannot harm others. Smokers' understanding is necessary" (FGD 307) |
| | SFH 2—Self-understanding is essential. If you look at a cigarette packet, there is a warning sign even though people smoke. Why? Because they do not want to change. They know. But they are not motivated". (FGD 105) |
| | SFH 3 - "No smoking inside the house is good for reducing exposure. If there is a specific place, sometimes people are too lazy to go to that place to smoke. So eventually, they do not smoke. But on the other side, they may start smoking in the house due to laziness. That's why I think if there is a specific place, the entire smoking product should keep there. So, if anyone wants to smoke, they have to go there for smoking". (FGD 303) |
| | SFH 4 - "Children don't follow advice. They follow the example. If you smoke and tell your children not to smoke, it does not make sense. It is the same if you want to make your house smoke-free; you need to set an example in front of your children. It would be best if you were strict about smoking inside the house". (FGD 305) |
| 2. Social norm and culture | SNC 1 - "Smoking is widespread in a social gathering. If you invite your relatives to your house, you will find a section of your relatives smoking and gossiping. Social Smoking is common practice. It is difficult to reduce exposure if a person doesn't understand what harm he is causing". (FGD 207) |
| | SNC 2 –"Woman cannot control a man, and in the case of smoking, it is almost impossible. The man never listens to us. They come up us hundreds of excuses for smoking. If we force them to quit, they get angry". (FGD 309) |
| | SNC 3 - "Woman empowerments help to break this norm. 80% loan takers from us are a woman. They told us that now their husbands give more value to their opinion. Now they listen to them". (FGD 103) |
| 3. Peer support | PS 1 - "I think peer support is the main medium. The spouse should be strict. If the husband respects her opinion, both can make the environment exposure-free. But in a rural area, it is difficult for us. We do not have that much freedom. If the grandparent is a smoker, it will be more difficult. In that case, the child's father needs to be more aware". (FGD 201) |
| | PS 2 - "If a child gets knowledge of smoking and ETS, in my opinion, they can motivate their parents. If the school teacher can deliver the message properly, they can influence their parents to stop smoking". (FGD 302) |
| | PS 3 –"Those who do not smoke should help the smoker. They should tell them the benefit. How healthy their life. It may affect the smoker". (FGD 203) |
| 4. Social awareness | SW 1 - "Why don't people use their knowledge? The only thing is they do not have any awareness. They know smoking cause cancer. Not that they do not have any idea about this matter. Awareness is necessary". (FGD, 206) |
| | SW 2 - "Education can create awareness. Children should learn about the harm of smoking. The schoolteacher should include an emphasis on this. Not only that, but no cigarettes should also sell beside school; otherwise, there will be no benefit". (FGD 101) |
| | SW 3 - "Seeing is believing. Those affected by smoking-related diseases like lung cancer or children suffering from respiratory problems due to secondary exposure to smoking can be an example for smokers. If it is possible to take the smoker to suffer persons, I think it will be more beneficial. It will work more adequately if the distressed person is close to the smoker". |
| | SW 4 - "Doctor, SACMO can help. I think for our area SACMO is more appropriate. Many people go to them for basic treatment. They also have a good understanding of the local people. If they talk about passive smoking, I think it will help. Mainly if they can inform the woman, they can get the knowledge and tell their husband not to smoke in front of children". (FGD 106) |
| 5. Religious practice | RP 1 - "Religious practice is more important than knowledge and awareness. Most of the smokers in our locality are male. If they know it is not permissible to enter masjid after smocking, they may change their smoking concept. They will understand smoking is a sin and will stop smoking and reduce exposure". (FGD 207) |
| | RP 2 - "We reduce smoke during Ramadan. Due to fasting, we were not able to smoke during the daytime. The month of Ramadan is the best time to quit smoking. It will also help make a house smoke free". (FGD 303) |

a significant impact on reducing smoking exposure (Quotes PS 2). Those who do not smoke and close friends of the smoker can also significantly reduce smoking exposure. Those who do not smoke can be a living example of a healthy family. On the other hand, non-smoker friends can help the smoker friend understand the devastating effect of passive smoking on the smoker's family (Quotes PS 3).

## D. Social awareness

ETS exposure reduction can be achieved through social awareness. It was found that people know about ETS. However, they are not aware of the harmful consequence (Quotes SW 1). Children's awareness is critical to reducing exposure. Children can help their parents to make the house smoke-free. If they know the harmful effect of ETS, they may prevent themselves from exposure (Quotes SW 2). One of the most excellent ways to educate people; is by taking help from those already suffering from smoking or exposure to tobacco. By seeing their condition, smoke will be mindful of exposing others (Quotes SW 3). The health professional should reduce exposure, mainly Sub-Assistant Community Medical Officers (SACMO) and have a good bonding with the local people. Local people took help from them even out of office time as they live in the locality. They have a strong influence on local people (Quotes SW 4).

## E. Religious practice

Religion has an integral part of most humans. A person's belief is very much influenced by religion. Faith can help a person drop an unhealthy mindset and take a healthy one. As most of the population in Bangladesh is Muslim, Islamic law and Islamic lifestyle are reflected by participants (Quotes SP 1). Islamic rules also help to make the house smoke-free. During Ramadan, Muslims do not eat anything from sunrise to sunset. So, no one can smoke in the daytime. If this practice continued, it could help stop smoking, thus helping to reduce ETS (Quotes SP 1).

## Phase 3—Evaluation of the model

The survey described five core activities that improved religious practice, implemented a smoke-free household, inspired by a peer, increased social awareness, and influenced social norms and culture. The core activities had 18 items, and out of them, seven items had achieved consensus in the first round of the modified Delphi technique. The items were 1. Putting up a no-smoking sign made by a child of the house on the doorway, 2. Seeking help from those who already practice smoke-free rules in the household,3. Changing clothes and hand washing before going near a child after smoking, 4. Practicing religious rules properly, 5. Educating children about the impact of ETS, 6. Distributing leaflets, banners, and stickers about ETS and 7. Adapting to social value.

However, the 11 items that did not get consensus in the first round were used for the second round. Among the six items achieved consensus; 1. Religious leaders should act as educators to inform the local people about the harmful effects of smoking and ETS, 2. Support from friends and families, 3. Engaging local social welfare club members to disseminate the destructive impact, 4. Using signboard and poster on ETS at local tea stalls, 5. Organizing friendly sports with an embedded message of ETS awareness and 6. Woman empowerment

A total of 13 items met ≥75% of all the things consensus criteria (Table 4). Finally, a harm reduction model was developed, comprising core activities and the elements (Fig 2).

Table 4. Descriptive statistics and interquartile range of 1$^{st}$ and 2$^{nd}$ round.

| Items | 1$^{st}$ Round (n = 21) | | | | 2$^{nd}$ Round (n = 19) | | | |
|---|---|---|---|---|---|---|---|---|
| | Mean | SD | Median | IQR | Mean | SD | Median | IQR |
| **Implement Smoke-free Household** | | | | | | | | |
| Specifying a place for smoking in the household | 6.71 | 3.01 | 8 | 4 | 7.0 | 2.86 | 8.50 | 3.25 |
| Design a no-smoking sign by a child of the house and put it on the doorway. | 8.47 | 0.74 | 9 | 1* | - | - | - | - |
| Using a calendar and mark it after every smoking inside the house and check at the end of every month. | 6.61 | 2.15 | 7 | 4 | 6.77 | 2.04 | 7.50 | 4 |
| Taking help from those who already practice smoke-free rules. | 8.38 | 0.92 | 9 | 1* | - | - | - | - |
| Developing a habit of smoking before coming home. | 6.80 | 3.09 | 8 | 4 | 6.50 | 3.24 | 8 | 6.25 |
| Change dress and wash hands before going near a child after smoking. | 8.52 | 0.67 | 9 | 1* | - | - | - | - |
| **Improve Religious practice** | | | | | | | | |
| Practicing religious rules properly reduces ETS. | 8.57 | 0.50 | 9 | 1* | - | - | - | - |
| Engaging religious leaders to inform the local people about the effect of smoking and ETS. | 8.04 | 1.28 | 9 | 2 | 8.5 | 0.70 | 9 | 1* |
| **Inspire from Peer** | | | | | | | | |
| Support from spouse, parents, and other household members. | 8.23 | 1.59 | 9 | 1.50 | 8.5 | 0.92 | 9 | 1* |
| Educating children about the impact of ETS exposure | 8.42 | 1.39 | 9 | 0.50* | - | - | - | - |
| **Increase Social Awareness** | | | | | | | | |
| Engaging members of local social welfare club to aware local people about ETS. | 8.19 | 1.07 | 9 | 2 | 8.4 | 0.78 | 9 | 1* |
| 'Social movement' such as implementing a fine for smoking inside the house by local social welfare club. The amount will be used for child welfare in the locality. | 6.52 | 2.76 | 7 | 5 | 6.27 | 2.90 | 7 | 5.25 |
| Increasing social awareness through the seminar. | 7.42 | 2.15 | 9 | 3 | 7.27 | 2.27 | 9.5 | 3.25 |
| Distributing leaflets, banners, and stickers about ETS. | 8.28 | 1.10 | 9 | 1* | - | - | - | - |
| Using signboard and poster on ETS in the local tea stalls. | 7.90 | 1.51 | 8 | 1.50 | 8.38 | 0.77 | 9 | 1* |
| Organizing friendly sports with the embedded message of ETS awareness. | 7.95 | 1.43 | 9 | 2 | 8.44 | 0.78 | 9 | 1* |
| **Influence of Social Norm and Culture** | | | | | | | | |
| Woman empowerment | 8.09 | 0.88 | 8 | 1.50 | 8.27 | 0.75 | 8 | 1* |
| Adapting social value | 8.33 | 0.85 | 9 | 1* | - | - | - | - |

*Achieved consensus

## Discussion

This research is among the first research to reduce ETS exposure among children in Bangladesh's rural area. The study identified five core activities through a modified Delphi technique, reducing ETS exposure among children in the household. The activities were listed as improving religious practice, implementing smoke-free homes, inspiring a peer, increasing social awareness, and influencing social norms and culture.

Religion has a significant influence on human life. It also influences human behaviour and social culture. As Bangladesh is a Muslim country where most of the population is from the Islamic faith, the belief and practice of the religion and the religious leaders' support were found an effective way to reduce the ETS exposure as smoking is prohibited according to the Islamic rules [21]. This finding is similar to a recent study conducted in Bangladesh, where it was proved that religious leaders greatly influence general people and, utilizing them, positively can effectively reduce ETS [22].

Implementing smoke-free households has been a widely accepted practice in developed and developing countries for a long time to reduce ETS exposure. However, In Bangladesh, particularly in rural areas, it is difficult to make a house smoke-free. No smoking policy is practiced more in urban areas than in rural areas [23]. The household structure, surrounding environment, and rural area behaviour are barriers to making the house smoke-free. However, although it seems complicated, this research found that different initiatives involving children

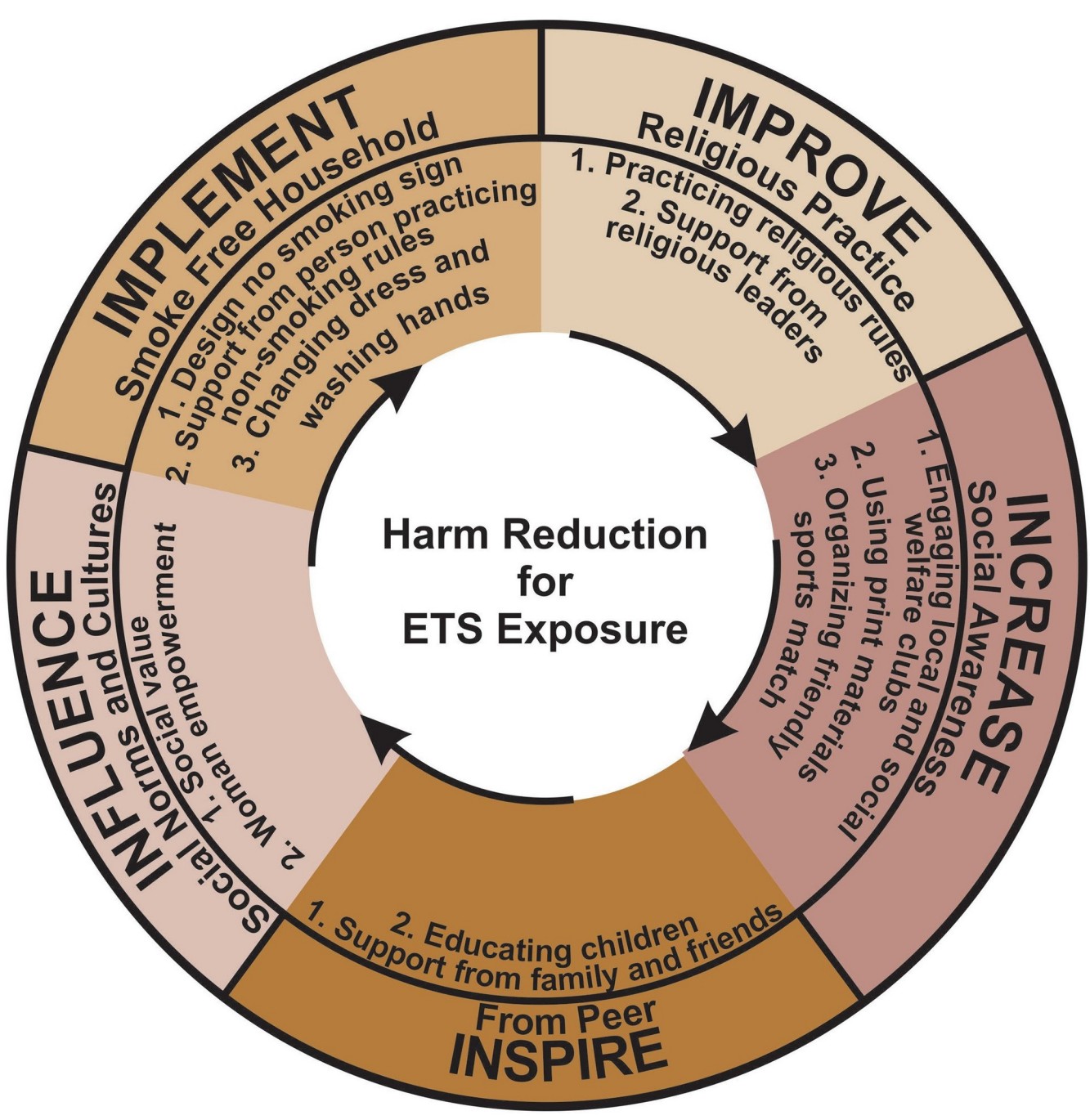

**Fig 2. Harm reduction model for ETS exposure.**

to make the home smoke-free, practicing healthy behaviour, and peer support can reduce ETS exposure in the household. Similar results were found in previous research in an urban area of Bangladesh [24]. The result is also consistent with the finding of a study conducted in China [25].

Peer support is significant to change any behaviour as the peer primarily influences a household and outside the home. Close family members and friends can help to improve and adopt a practice. Furthermore, children are considered as most precious to any parents.

Children's support also can change smokers' behaviour to reduce exposure. This study's finding also compliments the result from a similar survey conducted in China, where family interaction was found to reduce ETS exposure, and another from Ghana, where friends and spouses played a substantial role in reducing exposure of ETS [26, 27].

Awareness is essential to drop an unhealthy behaviour. Increased individual awareness can help understand health behaviour and social knowledge and motivate people to adopt healthy practices. Different initiatives, such as using print media items, seminars, workshops, and friendly sports, can help people become aware of healthy behaviour. Similar results also observed in the United States among African-born Women indicate that attending seminars and workshops, using media, and appealing to community members' priorities, including protecting their children, can reduce ETS exposure [28].

Every society has its norm and culture, which are considered a guideline of behaviour for the local people. The study found that social value, and woman empowerment had influenced the exposure. In Bangladesh, people do not smoke in front of the elderly out of respect, which reduces exposure. Alternatively, this same attitude can help reduce exposure by showing care for children. Moreover, a restricted cultural norm has limited freedom to ask her husband not to smoke inside the house. By breaking the social taboo on a woman, giving the proper right to them can also effectively reduce smoking exposure. The finding aligns with the qualitative study conducted among rural Bangladeshi and urban Indian women who found male attitudes significantly reduced smoking [29]. Moreover, Bangladesh's cultural norm gives men superior social status, allowing them to smoke in the house increasing ETS exposure inside the household [9].

The study had some limitations. Due to the cultural aspect of the rural area, most of the respondents were male, which may lead to selection bias. The cross-sectional study design measured both independent and dependent variables in a single point of time. All variables were self-reported, leading to misclassification due to recall and reporting bias. The outcome did not validate with cotinine estimation. The reason is that the Bangladeshi people have tobacco smoking and habit of betel chewing with tobacco which can be a mixed-use of tobacco and can bias the result with cotinine estimation due to the misclassification of tobacco use (inclusive of without smoke). Additionally, the study only focused on household factors. Other settings, such as restaurants, workplaces, or public places, were not included in the study, whereas these are potential places for ETS. Furthermore, only one district of Bangladesh had been included which may cause the result differ from other districts.

Despite having limitations, the research has its strength. The study used more than one method to identify and measure the correct variables for analysis. The mixed-method approach helps to complete the objective of the research comprehensively. As smoking is considered a sensitive topic, only conducting an interview means little opportunity to find the responders' actual behaviour patterns and families. However, the interview gave the essential information to form more quantitative questions that provide more information about household practice with ETS exposure, especially related to the children. Over time, several interviews provided a more detailed description and a greater understanding of ETS's family views. Nevertheless, a large randomized control trial to detect the model's effectiveness in the long term using biochemical validation of children is recommended for future research.

## Supporting information

**S1 File. Questionnaire for cross sectional study.**
(PDF)

**S2 File. Qualitative questions.**
(PDF)

**S3 File. Questionnaire for modified Delphi method.**
(PDF)

# Acknowledgments

The first author acknowledges Naresuan University, Thailand to award him Naresuan University International Student Scholarship 2016 for pursuing Doctor of Public Health degree.

# Author Contributions

**Conceptualization:** Rishad Choudhury Robin, Narongsak Noosorn.

**Data curation:** Rishad Choudhury Robin.

**Formal analysis:** Rishad Choudhury Robin.

**Investigation:** Rishad Choudhury Robin.

**Methodology:** Rishad Choudhury Robin.

**Project administration:** Rishad Choudhury Robin, Narongsak Noosorn.

**Supervision:** Narongsak Noosorn.

**Writing – original draft:** Rishad Choudhury Robin.

**Writing – review & editing:** Rishad Choudhury Robin.

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
