## [Decision Letter · Decision Letter 0]

11 Jan 2022

PONE-D-21-24273A Harm Reduction Model for Environmental Tobacco Smoke Exposure among Bangladeshi Rural Household Children: A Modified Delphi Technique ApproachPLOS ONE

Dear Dr. Noosorn,

Thank you for submitting your manuscript to PLOS ONE. After careful consideration, we feel that it has merit but does not fully meet PLOS ONE’s publication criteria as it currently stands. Therefore, we invite you to submit a revised version of the manuscript that addresses the points raised during the review process.

We look forward to receiving your revised manuscript.

Kind regards,

Md Mosharaf Hossain

Academic Editor

PLOS ONE

Journal Requirements:

PONE-D-21-24273

"No"

Reviewers' comments:

Reviewer's Responses to Questions

**Comments to the Author**

1. Is the manuscript technically sound, and do the data support the conclusions?

Reviewer #1: Partly

Reviewer #2: Partly

Reviewer #3: No

2. Has the statistical analysis been performed appropriately and rigorously? 

Reviewer #1: No

Reviewer #2: Yes

Reviewer #3: No

3. Have the authors made all data underlying the findings in their manuscript fully available?

Reviewer #1: Yes

Reviewer #2: Yes

Reviewer #3: No

4. Is the manuscript presented in an intelligible fashion and written in standard English?

Reviewer #1: Yes

Reviewer #2: Yes

Reviewer #3: No

5. Review Comments to the Author

Reviewer #1: 1， In the cross-sectional study, it was indicated an overwhelming participant was male (70.7%) majority were married (85.6%) and completed from university (38.5%). Please check the data about the participant completed university education.

2, It seems that the research finding can not support the research goal.

3, The manuscript looks like a research report, rather than a academic paper.

4, The tables in the manuscript need to be format again.

5, The language need polish.

Reviewer #2: • Page 6: under Phase 3: evaluation of the model; It is suggested to mention what modification was done to the Delphi technique for better clarification of the modification in the method.

• Page 6: under Phase 3: evaluation of the model; Calculation of the consensus on Interquartile range (IQR) was not clarified enough and recommended to have a proper explanation on how IQR was calculated based on the response score and the reason for the cut point is set at less than/ equal to 1. (This point links the result table shown in Page 17).

• Page 21: “The outcomes did not validate with cotinine estimation” which needs proper explanation to mention. The reason is that the Bangladeshi people have tobacco smoking and habit of betel chewing with tobacco which can be a mixed-use of tobacco and can bias the result with cotinine estimation due to the misclassification of tobacco use (inclusive of without smoke).

• In figure 2: Just want to clarify that one item was not listed in the figure. Only 12 items out of 13 consensus criteria were reflected with one point omitting “distributing leaflets, banners and stickers about ETS”. Is that point covered under “using print materials” or missed to mention?

Reviewer #3: PONE-D-21-24273: statistical review

SUMMARY: This is a study of environmental tobacco smoke exposure among children in a district of Bangladesh. Both qualitative and quantitative research methods have been implemented, across the three phases summarized by Figure 1. My review will focus on the statistical analysis of phase 1, which essentially relies on a logistic regression analysis of a cross-sectional sample.

MAJOR ISSUES

1. The sample includes households of 6 villages in the Munshiganj district, randomly selected from 64 districts. Although I understand logistics constraints, a sample of villages drawn from different districts would have been a much more natural procedure and results could have been extended to the Bangladeshi population. Taking villages from one district leads to results that are instead much more limited. I think that the overall paper should be rephrased by assuming that the target population is the rural part of the Munshiganj district.

2. If I understood correctly (not clear from text, please clarify), the dependent variable in the logistic regressions of Table 2 is always smoke exposure, included as a self-reported binary variable. I'm quite a bit concerned about the dichotomous definition of exposure: what do 0 and 1 exactly mean here? How can we be sure that all the subjects give to 0 and 1 the same meaning? Without a clear-cut definition of the dependent variable, the results of table 2 are difficult to interpret.

3. Table 2 displays the results of a battery of logistic regressions, where some relevant covariates are included separately and then adjusted for confounders. This is not a standard approach: relevant covariates should be included simultaneously. I'd welcome a single, carefully selected (see major issue 4 below) logistic regression that includes the ORs of all the relevant covariates and the confounders.

4. Model checking is overlooked. Without asking for a full model diagnostic, the authors should at least provide some residual analysis that show the goodness of fit of the model.

SPECIFIC ISSUES

1. Although the English is generally correct, some sentences should be revised. Examples are: “The cross-sectional study observed”, “Through purposive sampling, the experts were selected”, “completed from university”, “It's not liked the town”…

2. The caption of Table 2 remarks that the ORs are adjusted for some confounders (age, gender, income, religious belief, knowledge, environmental impact, media influence, perception, attitude, intention, perceived behavior control). Only some of these variables are summarized by Table 1: what about the others? In addition, Table 1 should display the ORs associated with these confounders, too.

6. PLOS authors have the option to publish the peer review history of their article (what does this mean?). If published, this will include your full peer review and any attached files.

Reviewer #1: No

Reviewer #2: No

Reviewer #3: No

---

## [Author Response · Author response to Decision Letter 0]

25 Jan 2022

Manuscript ID: PONE-D-21-24273

Title: A Harm Reduction Model for Environmental Tobacco Smoke Exposure among Bangladeshi Rural Household Children: A Modified Delphi Technique Approach

Journal Requirements:

Response: Corrected accordingly.

Response: Included the Questionnaire. 

"No"

Response: The authors received no specific funding for this work.

Response:Data included in the the following DOI 

https://doi.org/10.5061/dryad.f7m0cfxz7

Reviewer 1:

Reviewer Comment 

1.In the cross-sectional study, it was indicated an overwhelming participant was male (70.7%) majority were married (85.6%) and completed from university (38.5%). Please check the data about the participant completed university education.

Response: We are thankful to the honourable reviewer for his comments. It was 37.5%.

2. It seems that the research finding can not support the research goal.

Response: Thank you for the comment. The research goal was to develop a harm reduction model by applying the modified Delphi technique, and finally, the model was developed in this research.

3. The manuscript looks like a research report, rather than a academic paper.

Response: We apricated the honourable reviewer comments. As the research is from the first author's doctoral thesis, the detailing of the manuscript is extensive, which may looks like a report. 

4. The tables in the manuscript need to be format again.

Response: Done accordingly to the Plos One Guideline

5. The language need polish.

Response: Done accordingly. 

Reviewer 2:

Reviewer Comment:

Page 6: under Phase 3: evaluation of the model; It is suggested to mention what modification was done to the Delphi technique for better clarification of the modification in the method.

Response: Thank you for the query. We have included the following information “ The modified Delphi technique is alike to the full Delphi in terms of technique. The major adjustment contains of beginning the procedure with a set of prudently selected items. These pre-selected items may be drawn from several sources. The key advantage of this modification to the Delphi is that it typically improves the initial round response rate and also provides a solid foundation in previously developed work” 

• Page 6: under Phase 3: evaluation of the model; Calculation of the consensus on Interquartile range (IQR) was not clarified enough and recommended to have a proper explanation on how IQR was calculated based on the response score and the reason for the cut point is set at less than/ equal to 1. (This point links the result table shown in Page 17).

Response: To calculate the strength of the consensus, Interquartile range (IQR) is used. We have used the reference of Persai D, Panda R, Kumar R, Mc Ewen A. A Delphi study for setting up tobacco research and practice network in India. Tobacco induced diseases. 2016;14:4-. and Lefkothea Giannarou EZ. Using Delphi technique to build consensus in practice. Int Journal of Business Science and Applied Management,. 2014;9(2).

• Page 21: “The outcomes did not validate with cotinine estimation” which needs proper explanation to mention. The reason is that the Bangladeshi people have tobacco smoking and habit of betel chewing with tobacco which can be a mixed-use of tobacco and can bias the result with cotinine estimation due to the misclassification of tobacco use (inclusive of without smoke).

Response: Thank you for the suggestion. We have included the suggested portion.

• In figure 2: Just want to clarify that one item was not listed in the figure. Only 12 items out of 13 consensus criteria were reflected with one point omitting “distributing leaflets, banners and stickers about ETS”. Is that point covered under “using print materials” or missed to mention?

Response: The honourable reviewer is absolutely right. It was included under distributing leaflets, banners and stickers about ETS.

Reviewer 3:

1. The sample includes households of 6 villages in the Munshiganj district, randomly selected from 64 districts. Although I understand logistics constraints, a sample of villages drawn from different districts would have been a much more natural procedure and results could have been extended to the Bangladeshi population. Taking villages from one district leads to results that are instead much more limited. I think that the overall paper should be rephrased by assuming that the target population is the rural part of the Munshiganj district.

Response: We apricated the honourable reviewer's view and partially agreed with his comments. Obviously, it would be great to obtain data from various districts of Bangladesh. However, that was very time consuming and needed funding. As Bangladesh is not a big country, rural areas are almost the same everywhere. That’s why the rural area reflects the other plain area except for the three hilly areas we mentioned in the research.

2. If I understood correctly (not clear from text, please clarify), the dependent variable in the logistic regressions of Table 2 is always smoke exposure, included as a self-reported binary variable. I'm quite a bit concerned about the dichotomous definition of exposure: what do 0 and 1 exactly mean here? How can we be sure that all the subjects give to 0 and 1 the same meaning? Without a clear-cut definition of the dependent variable, the results of table 2 are difficult to interpret.

Response: In this research, exposure to passive smoking indicates exposure to another person’s tobacco smoke in the household for at least 15 minutes daily for more than one day every week in the past 30 days. Though we have collected the data through a self-administered questionnaire, the researchers gave a brief description of how to fill the data, and if they failed to understand, they were provided with a toll-free number to discuss with the researcher.

3. Table 2 displays the results of a battery of logistic regressions, where some relevant covariates are included separately and then adjusted for confounders. This is not a standard approach: relevant covariates should be included simultaneously. I'd welcome a single, carefully selected (see major issue 4 below) logistic regression that includes the ORs of all the relevant covariates and the confounders.

Response: Thank you for the comments. The adjusted portion of the table was not included separately. It was included together. However, only the significant outcome was shown in the table. For better understanding, we include the full table here.

Logistic regression: Odds ratios (ORs) and 95% confidence intervals (CI) of demographic variables and independents variables with dependent variable

Variable B Sig. OR 95% CI for OR

Lower Upper

Variables which are significant in the regression model

Smoker in the household

Yes Reference 

No -5.146 .000* .006 .002 .021

Household smoking rules

No/low implementation Reference 

Highly implementation -5.222 .000* .005 .001 .058

Social norm and culture 

Low influence Reference 

Moderate influence -3.094 .009* .045 .004 .461

High influence -3.771 .001* .023 .002 .224

Subjective Norm

Negative Reference 

Neutral -3.734 .017* .024 .001 .510

Positive -3.548 .019* .029 .001 .561

Variables which are not significant in the regression model

Gender

Male Reference 

Female -.115 .885 .891 .189 4.210

Age in year

18-24 Reference 

25-44 -1.182 .152 .307 .061 1.545

45-64 .034 .973 1.034 .153 6.977

65+ 1.339 .506 3.816 .074 197.624

Marital status

Single Reference 

Married -.039 .948 .961 .292 3.162

Education

No formal schooling Reference 

Primary schooling .653 .620 1.921 .146 25.259

Secondary schooling .536 .693 1.709 .119 24.493

College -.482 .743 .618 .035 11.008

University -.668 .650 .512 .028 9.223

Employment

Unemployed Reference 

Service holder -.415 .622 .661 .127 3.434

Business .191 .837 1.210 .197 7.420

Agriculture .789 .372 2.201 .389 12.448

Religion

Islam Reference 

Other 1.083 .156 2.953 .661 13.203

Household income in BDT

<5000 Reference 

5000-10000 .114 .899 1.121 .193 6.495

10001-15000 .916 .300 2.500 .442 14.127

15001-20000 -.938 .325 .391 .061 2.532

20001-30000 -1.185 .229 .306 .044 2.108

>30000 1.466 .197 4.330 .467 40.167

Number of children in the household

1 Reference 

2-3 .632 .306 1.882 .561 6.313

4-5 -1.524 .100 .218 .036 1.336

6-7 1.728 .834 5.632 .000 56707685.132

Media

Low Reference 

High -1.949 .277 .142 .004 4.799

Household and Environment

Low Reference 

High -.125 .813 .883 .314 2.478

Knowledge

Low Reference 

Moderate -.894 .113 .409 .135 1.236

High -.186 .796 .830 .203 3.399

Self-belief

Low Reference 

High 4.391 .250 80.689 .045 144177.473

Religious belief

Low Reference 

Moderate 1.256 .326 3.511 .287 43.026

High .130 .902 1.139 .143 9.060

Social awareness

Low Reference 

High .493 .688 1.638 .148 18.170

Intention

Negative Reference 

Neutral 1.321 .725 3.749 .002 5903.248

Positive -1.593 .183 .203 .020 2.119

Attitude

Negative and Neutral Reference 

Positive -2.359 .087 .094 .006 1.410

Perceived behavior control

Negative Reference 

Neutral -.683 .490 .505 .073 3.512

Positive -.028 .977 .972 .142 6.641

4. Model checking is overlooked. Without asking for a full model diagnostic, the authors should at least provide some residual analysis that show the goodness of fit of the model.

Response: Thanks for the comments. The model was checked. The model predicted 91.7% of the correct estimate. The model chi-square (x2=371.792, d.f. = 40, p = 0.000), assessing goodness-of-fit, was significant, indicating that the independent variables are not related to the log odds of the dependent variable. The Hosmer and Lemeshow test was not significant (x2=7.481, d.f. = 8, p = 0.486), confirming the model's goodness of fit.

6. Although the English is generally correct, some sentences should be revised. Examples are: “The cross-sectional study observed”, “Through purposive sampling, the experts were selected”, “completed from university”, “It's not liked the town”…

Response: Thank you. We have tried to correct the sentence.

7. The caption of Table 2 remarks that the ORs are adjusted for some confounders (age, gender, income, religious belief, knowledge, environmental impact, media influence, perception, attitude, intention, perceived behavior control). Only some of these variables are summarized by Table 1: what about the others? In addition, Table 1 should display the ORs associated with these confounders, too.

Response: Thank you for the query. Kindly Check the response of question no 3.

---

## [Decision Letter · Decision Letter 1]

4 Jul 2022

PONE-D-21-24273R1A Harm Reduction Model for Environmental Tobacco Smoke Exposure among Bangladeshi Rural Household Children: A Modified Delphi Technique ApproachPLOS ONE

Dear Dr. Noosorn,

Thank you for submitting your manuscript to PLOS ONE. After careful consideration, we feel that it has merit but does not fully meet PLOS ONE’s publication criteria as it currently stands. Therefore, we invite you to submit a revised version of the manuscript that addresses the points raised during the review process.

Please address the remaining comments from reviewer 2.

We look forward to receiving your revised manuscript.

Kind regards,

Md Mosharaf Hossain

Academic Editor

PLOS ONE

and

Jamie Royle

Staff Editor

PLOS ONE

Journal Requirements:

Additional Editor Comments (if provided):

Reviewers' comments:

Reviewer's Responses to Questions

**Comments to the Author**

1. If the authors have adequately addressed your comments raised in a previous round of review and you feel that this manuscript is now acceptable for publication, you may indicate that here to bypass the “Comments to the Author” section, enter your conflict of interest statement in the “Confidential to Editor” section, and submit your "Accept" recommendation.

Reviewer #2: All comments have been addressed

Reviewer #3: (No Response)

2. Is the manuscript technically sound, and do the data support the conclusions?

Reviewer #2: Yes

Reviewer #3: Yes

3. Has the statistical analysis been performed appropriately and rigorously? 

Reviewer #2: Yes

Reviewer #3: No

4. Have the authors made all data underlying the findings in their manuscript fully available?

Reviewer #2: Yes

Reviewer #3: No

5. Is the manuscript presented in an intelligible fashion and written in standard English?

Reviewer #2: Yes

Reviewer #3: Yes

6. Review Comments to the Author

Reviewer #2: The authors addressed my question points comprehensively in the manuscript. Well done and thank you.

Reviewer #3: I'm generally satisfied with most of the work made by the authors in this revision. I just have some residual requests:

1) I still think that taking villages from one district coud lead to results that are difficult to extend to the general population. Although the authors disagree on this point, I would suggest to add this issue among the possbile limitations of the study.

2) Lines 245-248: "The model predicted 91.7% of the correct estimate. The model chi-square (x2=371.792, df. = 40, p = 0.000), assessing goodness-of-fit, was significant, indicating that the independent variables are not related to the log odds of the dependent variable." The meaning of these two sentences is not clear. I suggest to remove them. The Hosmer and Lemeshow test provides enough information about the model goodness of fit.

3) Data: I was not able to retrieve the data from Dryad, using the address the authors provided, doi:10.5061/dryad.f7m0cfxz7. Please provide a link that directly points to the dataset

7. PLOS authors have the option to publish the peer review history of their article (what does this mean?). If published, this will include your full peer review and any attached files.

Reviewer #2: **Yes: **Sun Tun

Reviewer #3: No

---

## [Author Response · Author response to Decision Letter 1]

17 Jul 2022

Manuscript ID: PONE-D-21-24273

Title: A Harm Reduction Model for Environmental Tobacco Smoke Exposure among Bangladeshi Rural Household Children: A Modified Delphi Technique Approach

Journal Requirements:

Response: Corrected accordingly.

Please check reference numbers 6, 7

Reviewer 3:

Reviewer Comment:

1. I still think that taking villages from one district coud lead to results that are difficult to extend to the general population. Although the authors disagree on this point, I would suggest to add this issue among the possbile limitations of the study

Response: Thank you for the suggestion. I have added the limitation on page 22 lines 355-356.

2. Lines 245-248: "The model predicted 91.7% of the correct estimate. The model chi-square (x2=371.792, df. = 40, p = 0.000), assessing goodness-of-fit, was significant, indicating that the independent variables are not related to the log odds of the dependent variable." The meaning of these two sentences is not clear. I suggest to remove them. The Hosmer and Lemeshow test provides enough information about the model goodness of fit.

Response: According to the advice of the reviewer, the lines were removed.

3) Data: I was not able to retrieve the data from Dryad, using the address the authors provided, doi:10.5061/dryad.f7m0cfxz7. Please provide a link that directly points to the dataset

Response: Thank you for the query. Please check https://datadryad.org/stash/share/kNWzQQ4egnoLlymWwJ2xmOPFi0mS0qIh9lghKdZFmsE

---

## [Decision Letter · Decision Letter 2]

7 Oct 2022

A Harm Reduction Model for Environmental Tobacco Smoke Exposure among Bangladeshi Rural Household Children: A Modified Delphi Technique Approach

PONE-D-21-24273R2

Dear Dr. Noosorn,

We’re pleased to inform you that your manuscript has been judged scientifically suitable for publication and will be formally accepted for publication once it meets all outstanding technical requirements.

Kind regards,

Muhammad Tayyab Sohail

Academic Editor

PLOS ONE

Additional Editor Comments (optional):

Reviewers' comments:

Reviewer's Responses to Questions

**Comments to the Author**

1. If the authors have adequately addressed your comments raised in a previous round of review and you feel that this manuscript is now acceptable for publication, you may indicate that here to bypass the “Comments to the Author” section, enter your conflict of interest statement in the “Confidential to Editor” section, and submit your "Accept" recommendation.

Reviewer #2: All comments have been addressed

Reviewer #3: All comments have been addressed

2. Is the manuscript technically sound, and do the data support the conclusions?

Reviewer #2: Yes

Reviewer #3: (No Response)

3. Has the statistical analysis been performed appropriately and rigorously? 

Reviewer #2: Yes

Reviewer #3: (No Response)

4. Have the authors made all data underlying the findings in their manuscript fully available?

Reviewer #2: Yes

Reviewer #3: (No Response)

5. Is the manuscript presented in an intelligible fashion and written in standard English?

Reviewer #2: Yes

Reviewer #3: (No Response)

6. Review Comments to the Author

Reviewer #2: (No Response)

Reviewer #3: (No Response)

7. PLOS authors have the option to publish the peer review history of their article (what does this mean?). If published, this will include your full peer review and any attached files.

Reviewer #2: **Yes: **Sun Tun

Reviewer #3: No

---

## [Editor Report · Acceptance letter]

7 Feb 2023

PONE-D-21-24273R2 

A harm reduction model for environmental tobacco smoke exposure among Bangladeshi rural household children: a modified Delphi technique approach 

Dear Dr. Noosorn:

I'm pleased to inform you that your manuscript has been deemed suitable for publication in PLOS ONE. Congratulations! Your manuscript is now with our production department. 

Kind regards, 

on behalf of

Dr. Muhammad Tayyab Sohail 

Academic Editor

PLOS ONE